# Phytotherapy in Integrative Oncology—An Update of Promising Treatment Options

**DOI:** 10.3390/molecules27103209

**Published:** 2022-05-17

**Authors:** Amy M. Zimmermann-Klemd, Jakob K. Reinhardt, Moritz Winker, Carsten Gründemann

**Affiliations:** 1Translational Complementary Medicine, Department of Pharmaceutical Sciences, University of Basel, Klingelbergstrasse 80, CH-4056 Basel, Switzerland; amy.klemd@unibas.ch (A.M.Z.-K.); moritz.winker@unibas.ch (M.W.); 2Pharmaceutical Biology, Department of Pharmaceutical Sciences, University of Basel, Klingelbergstrasse 50, CH-4056 Basel, Switzerland; jakob.reinhardt@unibas.ch

**Keywords:** integrative oncology, phytotherapy, anticancer, natural products, medicinal plants, complementary medicine

## Abstract

Modern phytotherapy is part of today’s conventional evidence-based medicine and the use of phytopharmaceuticals in integrative oncology is becoming increasingly popular. Approximately 40% of users of such phytopharmaceuticals are tumour patients. The present review provides an overview of the most important plants and nature-based compounds used in integrative oncology and illustrates their pharmacological potential in preclinical and clinical settings. A selection of promising anti-tumour plants and ingredients was made on the basis of scientific evidence and therapeutic practical relevance and included Boswellia, gingko, ginseng, ginger, and curcumin. In addition to these nominees, there is a large number of other interesting plants and plant ingredients that can be considered for the treatment of cancer diseases or for the treatment of tumour or tumour therapy-associated symptoms. Side effects and interactions are included in the discussion. However, with the regular and intended use of phytopharmaceuticals, the occurrence of adverse side effects is rather rare. Overall, the use of defined phytopharmaceuticals is recommended in the context of a rational integrative oncology approach.

## 1. Introduction

In 2020, cancer was one leading cause of death worldwide with almost 10 million deaths, showing that cancer is a serious problem [1]. The relevance of tumour disease is furthermore expected to increase, reflected by a prediction of the WHO that new cancer cases will rise by approximately 70% over the next 20 years [2]. Despite the achievements in modern medicine, the treatment of cancer is challenging as therapies such as surgery, radiotherapy, and/or systemic therapy (chemotherapy, hormonal treatments, targeted biological therapies) have a range of severe side effects and a limited efficiency [3,4,5,6]. In addition, resistance to therapy is increasing [7,8].

Herbal medicine is one of the oldest forms of therapy on our planet. In all parts of the world, independent forms of healing with plants have developed over the centuries, such as Ayurveda in India, Kampo medicine in Japan, Sa-sang in Korea, and traditional Chinese medicine (TCM) [9]. The discovery of morphine in 1804 was the starting signal for rational drug discovery from plants [10,11]. In the 1930s, there was a shift from crude extracts and partially purified natural products to pure compounds [12]. However, due to the rapid progress in the field of chemistry in the 20th and 21st centuries, which has promoted high-throughput screenings (HTS) of synthetic compound libraries for drug development [10], natural products increasingly lost importance [13]. However, the limited chemical diversity of synthetic compounds has led to fewer drug approvals and a return to natural product-based drug research [14], which is reflected in the analysis of Newman and Cragg, who postulate that one-third of the drugs approved by the Food and Drug Administration (FDA) between 1981 and 2014 were based on natural products [15]. Taxol (Paclitaxel), derived from the Pacific yew tree (*Taxus brevifolia*) and used for the treatment of breast, lung, ovarian cancer, and Kaposi’s sarcoma [16], and camptothecin (CPT) (as well as the derivates topotecan and irinotecan), isolated from the bark and trunk of the Chinese happy tree (*Camptotheca acuminata*) and used for the treatment of against colon, lung, ovarian, breast, liver, pancreas, and stomach cancers [17], are prominent examples of natural products that are now produced synthetically and used in tumour therapy.

Aside from the synthetically produced natural products, transcultural phytotherapy is becoming increasingly important all over the world and has long since found its way into medical practice and self-medication [10]. The pooled prevalence of a meta-analysis from 2021 indicated that 23% of adult cancer patients use herbal medicines [18]. Significantly more female cancer patients (27% pooled prevalence) than male cancer patients (17% pooled prevalence) used herbal medicines [18]. It should be noted that the male patients were all prostate cancer patients and most female patients had breast cancer. The figures, therefore, indicate the prevalence of herbal medicine users within a cancer type. Another study confirmed that the preference for using herbal medicine among female breast cancer patients is relatively high (57.7%, of the surveyed phytotherapy users suffered from breast cancer), followed by female genital (15%), digestive organs (12%), and male genital (10.6%) [19]. The highest pooled prevalence was detected in Africa (40%) and Asia (28%) and in general higher across low- and middle-income countries compared to high-income countries [18]. Presumably, a culturally conditioned attachment to naturopathy plays an important role here. Without a doubt, another reason for this is a lack of treatment availability in low-income countries. For example, 8.3% of the essential cancer therapies are not available in low-income countries. Of the essential therapies, 57.7% are available but only at full cost, which is not affordable for many people [20]. The relatively low cost is thus an enormous advantage of herbal medicines. Income also has a significant influence on people’s lifestyles, which is also highly relevant in relation to cancer. Studies have shown an enormous influence of lifestyle factors, such as nutrition, as well as the consumption of nicotine and alcohol, on the development of cancer, suggesting that 1/3 to 1/2 of cancers could be prevented [21,22]. Lifestyle also plays an important role in the long-term survival and quality of life of cancer patients [23]. Consequently, it is not surprising that the idea of contributing to one’s own healing is becoming increasingly important. In this context, phytotherapeutic (self)-medication can also be integrated by the patient into an overall healing concept, with the aim of improving the patient’s prognosis, ameliorating side effects, and alleviating the anxiety associated with cancer therapy. Consequently, the therapy with herbal preparations falsifies certain attention. 

Modern, rational phytotherapy is not alternative medicine but is part of today’s evidence-based conventional medicine as classic naturopathic treatment. It must be distinguished from other naturopathic methods, such as homeopathy and anthroposophical medicine, aroma or Bach flower therapy, which also use herbal preparations [24,25]. According to the national and European definition, phytotherapy is the ‘the science-based medicinal use of plants and preparations derived from them, in the treatment, alleviation and/or prevention of disease or injury, according to recognised standards of quality, safety and efficacy [26]. The therapeutic importance of phytotherapy lies mainly in the treatment of non-acute and chronic-functional diseases with the focus on outpatient departments and self-medication, but rarely in the clinical-inpatient area [24]. Based on its historical development, the rationale underlying traditional phytotherapy has developed over the course of time, the former being based on scientifically verifiable principles and the latter mainly based on traditionally transmitted empirical medicine [27]. Generally, treatment with herbal medicines is perceived as natural and thus gentle and harmless [28]. Herbal preparations are usually mixtures of many substances, which actually leads to a relatively low side effect profile, as the individual components may be present in harmless concentrations and the effect is based on synergic effects [29]. Nevertheless, side effects may also occur. In addition, the interaction with conventional therapies involves some risks, such as the reduction of therapy efficiency [30,31]. Furthermore, herbal preparations are often characterised by low bioavailability, which can be reduced by semi-synthetic or synthetic analogues of natural products [32]. The combination of natural substances can sometimes also drastically increase the bioavailability of natural substances. For example, piperine can increase the bioavailability of curcumin by 2000% [33].

Phytotherapy drugs are known as phytopharmaceuticals and are defined in the European Union (EU) as Herbal Medicinal Products (HMPs). However, this does not include medicinal products with isolated plant ingredients or simulated synthetic natural substances. Phytopharmaceuticals are clearly differentiated from dietary supplements and functional foods [24]. 

Monographs demonstrating the efficacy and safety of herbal medicines provide the basis for national regulatory authorities. While the consideration of monographs is not binding, their disregard should be well justified [34]. At the EU level, these are the monographs of the European Scientific Cooperative on Phytotherapy (ESCOP) Commission or the Committee on Herbal Medicinal Products (HPMC) as a permanent body of the European Medicines Agency (EMA) [34,35]. The structure of the ESCOP monographs is based on the Summary of Product characteristics guideline (SmPC) and serves to illustrate the efficacy and safety of herbal medicinal products [35]. The HPMC monographs distinguish between ‘well-established use’ [36] and ‘traditional use’ and also correspond to an SmPC in their structure [34]. At the global level, the WHO Department of Traditional Medicine is developing monographs on the efficacy, safety, and quality of herbal medicines. In addition, information on dosage forms is included [37]. The WHO monographs have no official status but are merely recommendations.

As rational phytotherapy becomes increasingly important in the context of tumour disease, both among the general population and regulatory authorities, it deserves some attention and will be explored in more detail in this review.

## 2. Search Strategy

For the present review, we applied a predefined search strategy and included studies published in old herbal books and dispensaries and the PubMed database up to April 2022. The following key words were applied in the PubMed database as search terms using Boolean operators phytotharapy[tiab] AND (cancer[tiab] OR oncology[tiab]), phytotharapy[tiab] AND (cancer[tiab] OR integrative oncology[tiab]), boswellia[tiab] AND (cancer[tiab] OR oncology[tiab]), viscum album [tiab] AND (cancer[tiab] OR oncology[tiab]), ginkgo[tiab] AND (cancer[tiab] OR oncology[tiab]), ginseng[tiab] AND (cancer[tiab] OR oncology[tiab]), zingiber[tiab] AND (cancer[tiab] OR oncology[tiab]), curcumin[tiab] AND (cancer[tiab] OR oncology[tiab]). Only sources and publications in German or English language were included. Clearly irrelevant studies were eliminated by screening all the titles and abstracts of the publications identified through the conducted database searches. The remaining papers were evaluated by reviewing their full-text versions. In addition, the reference lists of all eligible papers were manually checked to minimise the risk of omitting relevant studies. Searches were conducted and screened according to the following selection criteria, which were determined by one review author (AZK): included objectives, analytical/chemical and biological method tested, outcomes reported, journal/impact, country of study/studies, and results. Chemical and biological data were extracted by one reviewer (AZK). The extracted data were reviewed by a second (chemical data: JKR) and a third reviewer (chemical and biological data: CG). Disagreements were resolved through discussions and consensus. Findings from the included publications were synthesised using chemical structures and a narrative summary.

## 3. Plants and Their Ingredients in Oncology

Currently, over 25,000 biologically active phytochemicals have been identified and have attracted the interest of research and development of new cancer therapies and drugs [38]. The intake of plant extracts, preparations, or plant ingredients is enjoying increasing popularity recently [39]. A selection of promising anti-tumour preclinical and clinical effects of plants and plant ingredients is described in the following pages. The selection was made on the basis of scientific evidence and therapeutic practical relevance, which does not claim to be complete.

In addition to the candidates described below with clear scientific evidence, there are a large number of other interesting plants and plant ingredients that can be considered for the treatment of cancer diseases or for the treatment of tumour or tumour therapy-associated symptoms. Over a thousand plants with such potential have now been identified [40]. The boundary to food is also blurred.

These include pomegranate (*Punica granatum*), grapefruit (Citrus × paradisi), and linseed (*Linum usitatissimum* L.), which are, for example, foods of phytotherapeutic interest [41,42,43]. Lycopene, which occurs abundantly in tomatoes (*Solanum lycopersicum*), resveratrol found in red wine and berries, and epigallocatechin gallate from green tea (*Camellia sinesnis*) are secondary plant substances that can play a therapeutic role in the nutrition of cancer patients (Figure 1) [44,45,46,47]. There is a strong consensus for the use of these plants and plant ingredients; however, currently, there is an absence of a sufficient number of clinical studies to conclusively confirm the effect in practice.

Last but not least, there are certain plants that do not have a direct effect on cancer cells, but which are nevertheless of great importance for accompanying therapy, as they have shown positive effects on tumour-associated side effects or the side effects of radiation and chemotherapy. A few examples are real aloe (*Aloe vera*), guarana (*Paullinia cupana*), valerian (*Valeriana officinalis* L.), and St. John’s wort (*Hypericum perforatum*). Real aloe is mainly used to treat the side effects of radiation therapy. However, clinical studies have indicated that real aloe does not contribute to the prevention of radiodermatitis in clinical studies [48,49,50,51]. Currently, there is an insufficient number of clinical studies available for the treatment of other radiation- or chemotherapy-associated side effects, such as stomatitis or proctitis [52,53]. 

Further, the high caffeine content of guarana and the associated cardiac and CNS stimulating effect makes guarana an interesting candidate for the treatment of tumour-associated or tumour therapy-associated symptoms of fatigue (Figure 1). Unfortunately, no benefit of guarana has been demonstrated in clinical studies in this regard [54,55,56,57]. Valerian is said to help cancer patients sleep through the night. Unfortunately, these effects could not be confirmed in clinical studies either [58]. Further, St. John’s wort is a preparation used to treat mild to moderate depression, because of the OATP2B1 inhibitory effect of the compound hypericin [59]. The effect of St. John’s wort is comparable to the effect of conventional antidepressant preparations, but there are no special clinical studies that have investigated the effectiveness of this plant in tumour patients [60].

## 4. Boswellia/Frankincense

Boswellia resin is a complex mixture of triterpenoids, mono-, sesqui-, and diterpenoids, as well as polysaccharides, and is also known colloquially as Frankincense [61,62]. Frankincense is used in various cultures for religious purposes and for the treatment of various diseases [63]. Different types of Boswellia have been used in TCM as well as in Ayurvedic medicine. In Western medicine, frankincense is mainly used for inflammatory diseases (inhibition of 5-lipooxygenase and, thus, leukotriene biosynthesis), as well as for asthma. Boswellia dry extract is generally well tolerated and available as a medicinal substance or as a dietary supplement [64,65].

Various preclinical studies have shown an anti-tumour effect of Boswellia in various cancer entities. The underlying mechanisms are complex. Boswellia inhibits various cancer-associated signal transduction pathways, such as Signal transducer and activator of transcription 3 (STAT3), WNT/β-catenin, AKT, extracellular signal-regulated kinases 1/2 (ERK1/2), nuclear factor ‘kappa-light-chain-enhancer’ of activated B-cells (NF-κB), nuclear factor of kappa light polypeptide gene enhancer in B-cells inhibitor (IκB), mammalian target of rapamycin (mTOR), and c-Myc [66]. Boswellia also influences both proliferation and the cell cycle by modulating the proteins involved [66,67,68]. It is known that chronic inflammation, associated with oxidative stress, promotes the development of cancer through DNA damage and promotes the proliferation of malignant cells during tumourigenesis [69]. Treatment with Boswellia can lead to a reduction in oxidative stress and counteract carcinogenesis [70,71,72]. Last but not least, Boswellia can lower pro-carcinogenic signals, like inflammatory signals, and thus be used as preventive tumour therapy [66]. It is known that 11-keto-β-boswellic acid (KBA), and 3-*O*-acetyl-11-keto-β-boswellic acid (AKBA) are mainly responsible for anti-tumour activity of Boswellia as they are highly competent in altering the inflammatory NF-κB and Akt pathway (Figure 2) [73]. NF-κB and Akt are involved in the cancer progression, as well as in the induction of chemo and radio resistance in cancer cells [74,75,76]. Several cancer cell line studies support the anti-tumour potential of Boswellic acids [73].

The clinical studies on the anti-tumoral effects of Boswellia are currently still in their early stages. The ClinicalTrials.gov study register only contains two clinical studies that investigate the direct influence of Boswellia on tumour diseases (as of February 2022). The aim of the studies is to investigate the influence of Boswellia extract on breast cancer tumours and brain and central nervous system tumours. Otherwise, there are mainly studies that examine the effect of Boswellia on brain tumours, as conservative therapies often do not achieve much in this regard. One of the problems with this type of tumour is the formation of oedema due to radiation therapy. The oedema can be treated with glucocorticoids, but there is an increased risk of infection with *Pneumocystis carinii* [77]. The effect of Boswellia (4200 mg/day) on oedema formation in 44 patients with primary and secondary brain tumours was examined in a randomised, double-blind, placebo-controlled study [78]. The oedema volume in the verum group was significantly lower [78]. This effect could be verified in a double-blind study from 2011 to investigate the influence of Boswellia dry extract (400 mg/day–1200 mg/day for seven days) on the formation of oedema [79]. A third study examined the palliative treatment of children with brain tumours with a daily dose of 126 mg/kg of an extract enriched with 3-*O*-acetyl-11-keto-β-boswellic acid. In this study, a general improvement in health was recorded in 8 of 19 study participants [80]. When applied topically, Boswellia is also used to treat radiation therapy-induced dermatitis. However, there is a lack of clinical studies to actually prove the possible effects of Boswellia [81].

Conclusions: In laboratory studies, Boswellia inhibits tumour cells through various mechanisms of action. Unfortunately, there is currently a lack of clinical studies to confirm these anti-tumour effects. Moreover, the effects of Boswellia on radiation-induced oedema formation in brain tumour patients have been better investigated clinically, and the effects on the inhibition of cerebral oedema could also be confirmed.

## 5. Mistletoe (*Viscum album* L.)

The medicinal use of mistletoe extracts for the treatment of ulcers was first described by Leonard Fuchs in 1543 [82]. Even today, extracts of European mistletoe enjoy great popularity among tumour patients and are used by about one-third of patients [83,84]. Subcutaneous application of mistletoe is well tolerated and considered safe [85]. Standardised mistletoe preparations are always adjusted to a certain lectin content, but preparations differ depending on the host tree and the extraction method. The ingredients responsible for the effect are mistletoe lectins, which are glycoproteins, and viscotoxins, which are small polypeptides [86,87,88].

In vitro and in vivo, mistletoe lectins have been shown to induce caspase-dependent apoptosis of tumour cells and immune cells even at low doses [89]. Mechanistically, this induction is based on a modulation of the JAK-STAT signalling pathway [90]. In addition, an apoptosis induction by interaction of mistletoe lectins with DNA is discussed in [91,92]. Studies also show that mistletoe lectins can induce increased expression of Fas ligand in T cells. Consequently, apoptosis induction may be enhanced in Fas^+^ cells [93]. In addition to the apoptosis-inducing effect of mistletoe, its immunomodulating effect also contributes to its anti-tumour capacity. The mistle lectins (ML 1, ML 2, chitin-binding ML), viscotoxins, and oligo- and polysaccharides of mistletoe stimulate the cells of the innate and adaptive immune system in a variety of ways [94]. In general, treatment with mistletoe extracts leads to an increase in the leukocyte count [94]. The phagocytosis rate, as well as the cell cytotoxicity of the cells of the non-specific immune system can be increased by treatment with mistletoe extracts [95]. Thus, mistletoe lectins can stimulate the production of granulocyte/macrophage colony-stimulating factor (GM-CSF) by mononuclear cells in the peripheral blood, thus activating granulocytes and monocytes in the bone marrow [96]. Based on viscotoxin-dependent and viscotoxin-independent mechanisms, in vitro studies showed stimulation of granulocyte activity [97]. The activity (phagocytosis activity and cytokine release) of macrophages was also increased by mistletoe lectins [98]. Furthermore, studies show that viscotoxins and poly- and oligosaccharides contained in mistletoe increase the proliferation and cytotoxicity of natural killer cells (NK cells) [99,100]. Regarding the effects of mistletoe on the adaptive immune system, studies have shown an increased antigen presentation and an increased production of cytokines that stimulate the immune system [95]. Moreover, lymphocyte proliferation could be increased by treatment with mistletoe extracts [94]. Mistletoe lectins and viscotoxins are responsible for this effect [94]. In addition, mistletoe lectins can promote the maturation of dendritic cells (DCs), which can sometimes be inhibited by tumour cells, and thus stimulate lymphocytes [101,102]. Most recently, it has also been shown to modulate cytokines and thus selectively induce a Th1 immune response [102].

Clinical studies on the effect of mistletoe on mortality in tumour patients are inconclusive. On the one hand, there are studies showing a longer average survival in the mistletoe arm [103,104,105,106,107,108], on the other hand, there are studies that found no influence of mistletoe on the mortality of tumour patients [109,110,111,112,113,114,115]. The influence of mistletoe on progression-free survival of tumour patients has also been investigated in some studies. Here, no significant improvement could be found through treatment with mistletoe [107,111,116,117]. Mistletoe is also being considered for improving the quality of life in tumour patients. There are a number of studies investigating this effect of mistletoe using standardised questionnaires and analogue scales [107,109,116,118,119,120,121]. Although the results are not all congruent and some studies have methodological flaws, the consensus is that subcutaneous administration of mistletoe can nevertheless improve the quality of life of tumour patients.

Conclusions: The anti-tumour effect of mistletoe is mainly based on apoptosis induction and immunomodulation. There are several preclinical studies that investigate and describe the mechanism of action of mistletoe. Methodologically good clinical studies are also available. However, no significant effect of mistletoe on either mortality or morbidity could be found. Only for the use of mistletoe to improve the quality of life of tumour patients is there a consensus recommendation.

## 6. Gingko

The seeds and leaves of the ginkgo tree (*Ginkgo biloba*) have been used in TCM for over 2000 years. The seeds are mainly used to treat cough, asthma, enuresis, festering skin infections, and worm infections, while the leaves are used to alleviate memory loss and cognitive diseases as well as cardiac arrhythmias, cancer, diabetes, and thrombosis [122]. *Ginkgo biloba* is a complex mixture of numerous substances. This contains, among other things, flavanol glycosides—such as quercetin and kaempferol—and terpene lactones (special extract ‘EGb761’, 2003). In various formulations, *ginkgo biloba* is one of the bestselling dietary supplements worldwide. The standardised preparation EGb761 (24% flavanol glycosides, 6% terpene lactones) is sold in Europe [122].

Studies have shown that the anti-tumoral effect of ginkgo is partially based on its anti-oxidative properties [123]. It helps in the elimination of free radicals and, thus, enables the inhibition of the angiogenesis necessary for the formation of metastases [124]. Further, the expression of invasion-associated proteins has also been shown to be reduced and the metastasis-associated factor heat-shock protein 27 has been shown to be downregulated [125,126]. The Wnt/β-catenin-VEGF signal transduction pathway is also inhibited by an extract from the ginkgo seed coat [127]. In laboratory experiments, ginkgo was also found to influence the mRNA level of proteins involved in proliferation and apoptosis and, thus, disrupt the ratio of pro- and anti-inflammatory proteins of the Bcl-2 family [128]. In another study, ginkgo also promoted apoptosis by activating the caspase signalling pathway [129]. Further, the protein levels of important signal transduction proteins for proliferation and inflammatory processes—such as phospho-ERK1/2, NF-κB, matrix metallopeptidase 2 (MMP2), Stat3/janus kinase 2 (JAK2), and mTOR—were also modulated in laboratory studies by ginkgo extract or ginkgolic acid (GA) (Figure 3) [130,131,132]. The results of a new study demonstrate the anti-proliferative effects of a ginkgo biloba trypsin inhibitor in human and murine triple-negative breast cancer cells [123]. Most recently, preclinical studies have additionally revealed the anti-inflammatory effects of gingko. The individual ginkgo substances quercetin and kaempferol were found to inhibit the cyclooxygenase-2 (COX-2) promoter activity with and without tumour necrosis factor alpha (TNF-α) stimulation in colon cancer cells (Figure 3) [123]. A further study confirmed a proliferation inhibitory effect of quercetin and kaempferol in various oral cancer cell lines. This effect was due to apoptosis induction [133]. Ginkgolides A, B, and C also contribute to the anti-tumour effect of ginkgo through different mechanisms. Ginkgolide A reduced neointimal hyperplasia, a hyperproliferative state of vascular smooth muscle cells by decreasing Erk1/2 signal transduction [134]. Ginkgolide B induced autophagy in lung cancer cells and inhibited the NLR family pyrin domain containing 3 (NLRP3) inflammasome in one study. This is relevant in the context of cancer because inflammation promoted by the NLRP3 inflammasome can lead to cancer [135]. Moreover, ginkgolide B could decrease the invalidity of breast cancer by suppressing the translation of zinc finger E-Box binding homeobox 1 (ZEB1) protein [136]. Recently, anti-tumoral effects of ginkgolide c have also been demonstrated. Thus, ginkgolide C interfered with the Wnt/β-catenin signalling pathway and thus induced apoptosis in colon cancer cells [137]. A recent study shows that the sesquiterpenoid bilobalide from ginkgo is also promising for cancer therapy. Bilobalide induced nuclear damage and apoptosis, as well as cell cycle arrest in gastric cancer cells. The anti-tumour effect was confirmed in the rat model [138].

However, the status of current clinical research is unclear. Studies have provided evidence of lower ovarian cancer risk and positive effects in treating colon cancer with ginkgo extract or components from ginkgo such as ginkgolides A and B [139,140]. Another study was unable to confirm the protective effect of ginkgo leaf extract on the carcinogenesis of various cancer entities and could not detect significant differences in the incidence of cancer [141]. However, the study has methodological flaws, such as a rather short follow-up time for carcinogenesis with a median of only 6.1 years. 

It was also examined whether ginkgo can contribute to the prevention of cytotoxic-related cognitive impairments in breast cancer patients. Here, however, a difference between the placebo and ginkgo groups could not be determined either objectively or subjectively [142]. However, more studies are required to assess its therapeutic benefit.

Further, bleeding complications, due to reduced platelet aggregation as a result of the inhibition of the ‘platelet activating factor’ (PAF), are known to be possible side effects [143]. In addition, the International Agency for Research on Cancer classified ginkgo leaf extracts as a possible group 2B carcinogen [144,145]. The toxic effect of quercetin, which has been documented for numerous cell lines, is responsible for this effect [146].

Conclusions: Preclinical research provides promising results regarding the therapeutic benefits of ginkgo in the treatment of cancer. The results from clinical trials, on the other hand, are contradictory. Ginkgo has also been classified as a possible group 2B carcinogen and has potentially dangerous side effects.

## 7. Ginseng

Ginseng is an East Asian plant, the roots of which have been used in TCM for the treatment of numerous ailments and diseases for over 2000 years [147]. Today, both American ginseng (*Panax quinquefolius*) and Asian ginseng (*Panax ginseng*), which is often described as red ginseng in the literature and obtained through a special processing method, have medical relevance [147]. The red colour of the root is retained and three additional pharmacologically important ginsenosides are formed. Ginsenosides, which belong to the class of triterpene saponins, are the active substances in ginseng. The root of ginseng is used in various forms, such as powder, capsule, tablet, gel, and liquid extract [147].

Laboratory studies have shown a direct anti-tumour effect for both American and Asian ginseng [147]. Asian ginseng was able to induce apoptosis and cell cycle arrest and inhibit angiogenesis and lung metastasis in the mouse model [148,149]. Further, American ginseng induced mitochondrial damage in colon cancer cells and inhibited the proliferation of breast cancer cells by suppressing MAP kinases [150,151]. In addition, to a possible application for direct cancer therapy or prevention, the use of ginseng for the treatment of states of exhaustion in tumour patients can be considered. Preclinical studies indicate an anti-inflammatory and cortisol-modulating effect [152,153,154]. Thus, ginseng was found to counteract tumour-associated exhaustion, which is characterised by an increase in inflammatory cytokines and cortisone dysregulation. It has been shown that low toxic ginsenosides, based on protopanaxatriol, protopanaxadiol, oleanolic acid, and ocotillol as their most prominent scaffolds (Figure 4, a more exhaustive discussion on the substitution patterns of these saponins can be found in [155]) isolated from ginseng, possess antiangiogenic and anti-inflammatory effects [156]. Through their anti-inflammatory action, ginsenosides prevent genetic and epigenetic damage, as well as the activation of oncogenes [156]. In the context of anti-tumour effects, ginsenosides have recently attracted attention. A study shows that ginsenoside Rh2, extracted from ginseng, induces mitochondrial reactive oxygen species (ROS) production and promotes cell apoptosis of cervical cancer cells [157]. An inhibitory effect of Rh2 on heat shock protein 90 alpha (HSP90A), which is overexpressed in human hepatoma cells, was also detected [158]. Other studies confirmed the suppression of the proliferation of MCF-7 breast cancer cells by [159,160] and the inhibition of non-small cell lung cancer cells by Rh2 [161,162]. A red ginseng extract enriched with the ginsenoside Rg3 suppressed cell division of lung cancer cells and induced mitochondria-dependent apoptosis [163]. The ginsenosides Rk1 and Rg5 promoted apoptosis in human liver cancer cells, as well [164]. Furthermore, ginsenoside (20S)-protopanaxatriol inhibited Akt/mTor signalling pathway in triple-negative breast cancer (TNBC) cells, inducing non-protective autophagy. The anti-tumour effects could be confirmed in the xenograft mouse model [165]. Moreover, gintonin, a glycolipoprotein from Panax ginseng is a promising anti-tumoral candidate with several molecular targets [156]. 

Unfortunately, the direct anti-tumour effect of ginseng has thus far barely been supported by clinical studies. Only one epidemiological study on breast cancer patients indicated an improved quality of life and survival time [166]. However, the study has methodological shortcomings—neither the dose nor the type of ginseng was taken into account. The study situation is better for the use of ginseng in tumour-associated exhaustion. Two studies that support the preclinical studies can be cited here. A randomised study from 2017 reveals positive effects of ginseng (3000 mg non-standardised red ginseng over 60 days, in addition to chemotherapy) on the severity and frequency of stress in the everyday life of the tumour patients, and the occurrence of fatigue (as measured by the Fatigue Symptom Inventory, FSI) [167,168]; moreover, the depression and anxiety levels of the 60 lung cancer patients on whom this study was conducted also improved [168]. In a second placebo-controlled study, 282 patients with various cancer entities were treated with American ginseng in three doses (750 mg, 1000 mg, and 2000 mg) or were given a placebo. The patients in the group that was administered 2000 mg of ginseng showed significantly better values on the Brief Fatigue Inventory Scale (BFI), a scale for evaluating the strength and impairment caused by exhaustion in everyday life [169,170].

Conclusions: Preclinical research on the anti-tumour effects of ginseng provides various mechanisms of action and positive effects. However, a direct anti-tumour effect has not yet been confirmed in the clinic, as there are currently merely a few studies on this and the existing studies have methodological deficiencies.

## 8. Ginger

Ginger (*Zingiber officinale*) is mainly known as a common kitchen ingredient. In addition to its use as a hot kitchen spice, ginger root is also used worldwide to treat tumours [171]. The folk medical use of the tuber was initially not evidence-based. Today, however, there are a number of studies that prove the effectiveness of ginger against numerous cancer entities [171]. Ginger is available as a dietary supplement or as a medicine in powder form [172]. The preparations do not have a standardised drug-extract ratio. Ginger is a complex mixture of numerous substances, most of which can be assigned to gingerols, shogaols, zingiberen, or zingerons (Figure 5) [173]. According to laboratory results, the anti-tumoral effect of ginger is based on the activities of these four named classes of substances.

Ginger has a direct therapeutic benefit in the treatment of tumour diseases, as it can induce cell death in tumour cells. For example, 6-shogaol in ginger can downregulate the mRNA of cell cycle proteins and cyclin levels and, thus, cause cell cycle arrest [174]. Furthermore, it induces apoptosis by modulating the mRNA level of various apoptosis proteins [174]. In another study, 6-gingerol at a concentration >100 µM was found to induce apoptosis in LNCaP cells by increasing caspase expression [175]. Further, the gingerols and shogaols regulate the mRNA levels of numerous chemokines and cytokines and inhibit signal transduction pathways, such as the AKT, mTOR, or STAT signalling pathways [171,175,176,177]. Additionally, 8-gingerol and 10-gingerol, as well as 8-shogaol and 10-shogaol also show promising anti-tumoral properties. Cell cycle arrest in colorectal cancer cells was induced by 8-gingerol and it promoted their apoptosis. Furthermore, 8-gingerol interfered with cell migration by inhibiting the epidermal growth factor receptor [178]. For 10-gingerol, studies show apoptosis-inducing effects in TNBC cells, as well as through activation of the mitogen-activated protein kinase (MAPK) pathway in human colon cancer cells [179,180]. In a mouse model for TNBC, an increase in caspase-3 activity was detected after treatment with 10-gingerol [181]. The growth of ovarian cancer cells could also be suppressed by 10-gingerol. The reason for this was the induction of a G2 phase arrest [182]. The shogaols, 8-shogaol and 10-shogaol, also gave promising results. Treatment of human colon cancer cells with 8-shogaol or 10-shogaol resulted in reduced gene expression of B-cell lymphoma 2 (Bcl-2), and increased expression of p53 upregulated modulator of apoptosis (PUMA) and cleaved caspases 3 and 9. The shogaols thus induced apoptosis via the oxidative stress-mediated p53 pathway [183]. Moreover, tumour cells can prevent cell aging by shortening the telomeres by expressing the enzyme telomerase. Ginger can lower the activity of telomerase in lung cancer cells [184]. Another therapeutic benefit of the ginger tuber is the suppression of tumour invasion and metastasis by inhibiting matrix metalloproteinases, which can promote tumour invasion and metastasis due to their ability to degrade type IV collagen (important collagen of the basement membrane) [177,185,186,187].

In addition to the direct therapeutic anti-tumour effects of ginger, preventive effects are also described. The inhibition of COX-2 is one of the well-known anti-inflammatory effects of ginger [188,189]. COX-2 is overexpressed during carcinogenesis and is, therefore, probably associated with the development of cancer [190].

Unfortunately, the promising effects of preclinical research with regard to the therapeutic effectiveness of ginger root in tumour diseases have not yet been confirmed in clinical studies. The number of clinical studies examining the effect of ginger on the mortality of cancer patients is small, and the few studies that are conducted show methodological weaknesses. For example, at ClinicalTrials.gov (as of February 2022), there is only one randomised, controlled study that provides indications that ginger could reduce the proliferation of colon cancer cells and increase the rate of apoptosis in patients with an increased risk of colon cancer [191]. However, a methodological weakness of the study remains the small number of participants (*n* = 20). In a study, patients with ovarian cancer were treated with 2 g of ginger per day in addition to cytoreductive surgery and platinum-based chemotherapy. Six months after treatment, significantly fewer metastases were detected in the ginger group than in the placebo group [192].

In addition to its anti-tumour properties, ginger has proven to be a potent anti-emetic and, therefore, can be used in conjunction with tumour therapy (radiation therapy and chemotherapy) to reduce therapy-associated nausea [193,194].

Conclusions: Overall, the preclinical study results on the anti-tumour effects of ginger are interesting and promising. The clinical studies do not contradict the positive effects of ginger, but far more methodologically strong studies are necessary in order to be able to prove the therapeutic effectiveness of ginger to fight tumour diseases.

## 9. Curcumin

Curcumin is a secondary plant component of the turmeric plant (*Curcuma longa*). Turmeric has been used in TCM for various ailments for 2000 years. In 1937, turmeric found its way into modern medicine due to its anti-inflammatory properties [195]. Today, turmeric can be found in pharmacies and drugstores in powder form, as a liquid preparation, and in the form of capsules or tablets. In addition, the turmeric root is the main component of the curry powder often used in the kitchen. The amount of curcumin in the commercially available turmeric root varies between 0.6% and 6.5% of the dry weight [196]. This variance, as well as the poor bioavailability and chemical instability of curcumin (Figure 6), makes it difficult to present a general assessment of the effectiveness of the products [197]. Therefore, current galenics are attempting to significantly increase bioavailability through nanoemulsions/nanomicelles.

When describing the effect of curcumin on tumour diseases, the concentration is decisive. In small amounts, curcumin has an anti-oxidative effect, whereas high curcumin concentrations have a pro-oxidative effect [198]. According to laboratory results, curcumin could have preventive effects on carcinogens. For example, curcumin showed a strong effect against oxygen radicals [199]. Oxygen radicals play a role in the development of cancer, as they induce DNA strand breaks and can thus promote mutagenesis [200]. In addition, oxygen radicals influence redox-sensitive amino acid residues of various proteins and transcription factors, such as MAPK, phosphoinositide 3-kinases (PI3K)/Akt, activator protein 1 (AP-1), NFκB, STAT3, p53, and numerous others [201]. This influence disrupts cellular homeostasis and, thus, promotes the development of cancer. Last but not least, curcumin modulates membrane structure and permeability by activating lipid peroxidation [199,202,203]. Finally, mouse studies have shown that curcumin can increase the butyrate level and, on the one hand, inhibit the proliferation of the epithelial stem and precursor cells and, on the other hand, counteract inflammation and allergies [204,205]. Recent research shows anti-proliferative effects of curcumin on various breast cancer cells and papillary thyroid cancer cells [206,207]. Furthermore, curcumin promoted apoptosis in papillary thyroid cancer, as well as in colon carcinoma cells [206,208].

Unfortunately, the preclinical results of the anti-tumour activity of curcumin have barely been verified in clinical studies conducted thus far. In the database ClinicalTrial.gov, there are currently (as of February 2022) 56 studies that investigate the direct influence of curcumin (not in conjunction with other natural substances) on cancer. Of these studies, 30 have already been completed or terminated and the results of eight of these studies are available. Moreover, only one small study with 22 participants is published in a peer-review journal [209]. In the study, patients with familial adenomatous polyposis (FAP), an autosomal dominant precancerous condition with 100% colon carcinoma formation in young adulthood, were treated with 100% curcumin twice a day for 12 months. The treatment was generally well-tolerated, but no difference was found between the curcumin group and the placebo group [210]. Five other controlled studies dealt with the influence of curcumin on the progression and therapy-associated side effects of tumour diseases. One of these studies with two different target parameters examined the influence of curcumin on the survival and quality of life of prostate cancer patients [211]. With regard to survival, no difference could be determined between the verum and placebo groups, but there was an improvement in the quality of life in the turmeric group. Unfortunately, the study indicated methodological deficiencies, such as the lack of power analysis and a small sample size. The other three studies examine the influence of curcumin on the side effects of radiotherapy. Two of the three studies indicate positive curcumin effects [212,213]; the third study did not find any significant differences between the active and placebo groups. However, all three studies reveal a biased presentation due to small samples or methodological deficiencies.

Conclusions: Curcumin has been well studied in laboratory studies and its positive effects and mechanisms of action with regard to anti-tumour effects are well described. The status of clinical studies for curcumin is relatively good, but the positive effects from the preclinical stage have not yet been confirmed. Clinical studies have only shown that curcumin leads to an improvement in the side effects of radiotherapy. Unfortunately, the studies are generally rather small and often have methodological deficiencies.

## 10. Risks, Side Effects and Interactions

The use of phytopharmaceuticals is becoming increasingly popular among cancer patients. Approximately 40% of the users of phytopharmaceuticals are tumour patients [39]. However, consuming phytopharmaceuticals does not replace standard therapy with chemotherapeutic agents, immunomodulators, or radiotherapy. The combination of phytopharmaceuticals with standard therapeutic agents for treatment is complex and there are a few aspects that must be considered. One problem with standard medication is the low bioavailability of oncological drugs. In addition to the poor solubility in water, the rapid metabolism by the cytochrome 450 enzymes (CYP450 enzymes) in the liver also plays a role. The preparations in which the metabolism through CYP450 plays a role, for example, include Tamoxifen, Cyclophosphamide, Docetaxel, Paclitaxel, Ifosfamide, Irinotecan, Imatinib, Flutamide, Tegafur, Gefitinib, Etoposide, Teniposide, Thalidomide, and Vincristine, among others (Figure 7) [30]. Consequently, the inhibition of the CYP450 enzymes can improve the bioavailability and, thus, the efficiency of standard therapeutic agents. However, the narrow therapeutic range of many cancer drugs must also be considered. The slowdown in metabolism can lead to drug accumulation and toxic side effects. Problems can also arise here with prodrugs since an inhibitory influence on the metabolism can lead to the failure of the therapy. Further, induction of the CYP450 enzymes can also have negative effects if the therapeutic agent is metabolised so quickly that the therapeutic effects are limited [31]. For combination therapies, this implies that very precise examinations of the times and periods of drug intake are rather important in order to avoid undesirable side effects.

For Boswellia, an effect on inhibitory effect on CYP3A4/5 and CYP2C9 could be found in the microsome liver model [214]. However, these effects were significantly less potent in the physiological sandwich-cultured human hepatocytes model [214]. Nevertheless, caution is advised in combination with CYP2C9 sensitive substrates, like the breast cancer drug Tamoxifen [30,214]. In contrast, neither an inhibitory (inhibition above 50% of marker reaction) nor a stimulatory potential on clinically relevant CYP enzymes (CYP1A2, CYP2A6, CYP2B6, CYP2C8, CYP2C9, CYP2C19, CYP2D6, CYP2E1, and CYP3A4) could be detected for mistletoe [215]. There are studies that show that ginkgo extract modulates the CYP450 enzymes. However, there are also studies that show that ginkgo extract induces the synthesis of CYP2C19 and CYP2C9 and can, thus, lead to the faster metabolises of certain preparations like cyclophosphamide [122]. Thalidomides are also metabolised by CYP2C19 and the combination with ginkgo should therefore be evaluated carefully [216]. For ginseng, a number of in vitro studies show an influence on CYP enzymes, such as CYP3A4, CYP1A2, and CYP2D6 [217], but a relevant effect has not been confirmed in clinical studies [218]. Further, according to studies, ginger also inhibits the CYP450 enzymes [30,219,220,221]. For the ginger ingredients 6-, 8-, and 10-gingerol inhibitory effects on the CYP450 enzymes could be found in fluorescence assays [220]. The effect on CYP2D6 was rather weak, but CYP2C9 was strongly inhibited and the inhibition of CYP3A4 was also significant. Consequently, caution is advised in the case of combinatorial treatment with, for example, tamoxifen (CYP2C9 and CYP3A4 involvement), cyclophosphamide and etoposide (CYP3A4 involvement) [30,220]. Curcumin is also an inhibitor of CYP450 enzymes. The strongest effect occurred on CYP3A4 [222]. The CYP1A2, CYP2C9, and CYP2D6 enzymes were also inhibited in their activity [222]. Consequently, combination therapy of various standard chemotherapeutic agents, such as tamoxifen (CYP3A4, CYP2C9, CYP2D6 involvement), cyclophosphamide (CYP3A4 involvement), etoposide (CYP3A4, CYP1A2 involvement), flutamide and tegefur (CYP1A2 involvement), as well as gefitinib (CYP2D6 involvement) with curcumin must be thoroughly evaluated [30].

Further, certain standard cancer drugs are transported using P-glycoprotein, also known as ‘multidrug resistance protein’ 1 (MDR1) or ‘ATP-binding cassette sub-family B member’ 1 (ABCB1). The most important of these are vinlastin and vincristine, paclitaxel and docetaxel, doxorubicin and daunorubicin, topotecan and etoposide, and dasatinib and gefitinib [223]. Thus, the interaction of phytotherapeutic agents, such Boswellia [224], ginkgo [225], ginseng [226], ginger [227], and curcumin [228] with P-glycoprotein can also modulate the bioavailability of standard cancer drugs and lead to side effects [30,31].

In addition to the interaction of herbal preparations with the CYP450 enzymes or the P-glycoprotein transporters, direct interactions with other therapeutic agents are also possible. Red ginseng significantly increased the elimination half-life of 5-fluorouracil (5-FU) in the rat model and revealed synergistic effects with 5-FU on gastric cancer cells [229]. For curcumin, interactions with a number of standard preparations such as docetaxel, irinotecan, doxorubicin, idarubicin, cyclophosphamide, 5-FU, histone acetylase inhibitors as well as radiation and partial hepatectomy have been described [230,231,232,233,234,235,236,237,238,239,240,241]. In most cases, an inhibitory effect was observed through additional treatment with curcumin, but the dose and the time of the curcumin administration were of great significance.

In addition to the possible interactions with standard cancer therapeutics, the direct side effects of the herbal preparations still need to be considered. Treatment with Boswellia is characterised by a small number of mild side effects [242]. These include mainly gastrointestinal discomfort. Allergic reactions hardly occurred [242]. Mistletoe’s side effects consist mainly of flu-like symptoms, fever, and local injection site reactions [243]. In some cases, allergic reactions have been reported, as well as reversible liver toxicity after injection of high doses [243]. The most commonly reported side effects of ginseng are headache, hypertension, diarrhoea, insomnia, skin rash, and vaginal bleeding [244], and of ginkgo, stomach upset, headache, dizziness, constipation, rapid heartbeat, and allergic skin reactions [245]. For ginger, most side effects are related to the digestive tract. Thus, heartburn, nausea, diarrhea, and abdominal pain may occur [246]. In addition, cardiovascular and respiratory symptoms have been observed in patients treated with ginger after laparoscopic surgery [246]. Curcumin is known to cause headaches, rash, nausea, diarrhea, and yellow stools [247]. In addition, an increase in serum alkaline phosphatase and lactate dehydrogenase was observed [247].

## 11. Conclusions

The intake of phytopharmaceuticals is becoming increasingly popular recently, but these herbal medicines are usually not suitable for use in acute and emergency medicine or as the sole treatment for serious illnesses. The risks relate mainly to the use of herbal preparations of unsuitable quality, an incorrect assessment of their effectiveness, as well as a lack of knowledge of undesirable side effects and possible interactions among chemical-synthetic medicines, herbal medicines, and foods. However, with regular and intended use, the occurrence of side effects is rather rare and the use of defined phytopharmaceuticals in accompanying oncological therapy is recommended in the sense of rational integrative oncology treatment.

## Figures and Tables

**Figure 1 molecules-27-03209-f001:**
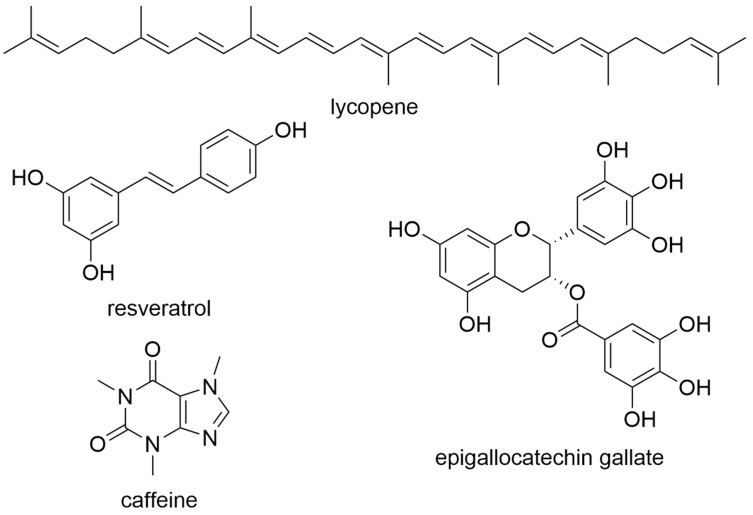
Natural products commonly found in food plants.

**Figure 2 molecules-27-03209-f002:**
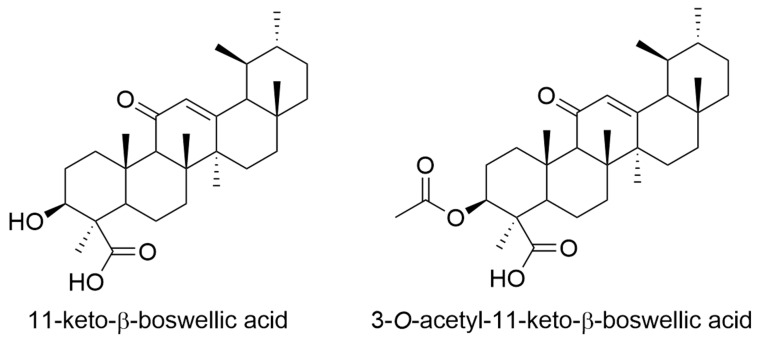
Examples of active constituents from *Boswellia* species.

**Figure 3 molecules-27-03209-f003:**
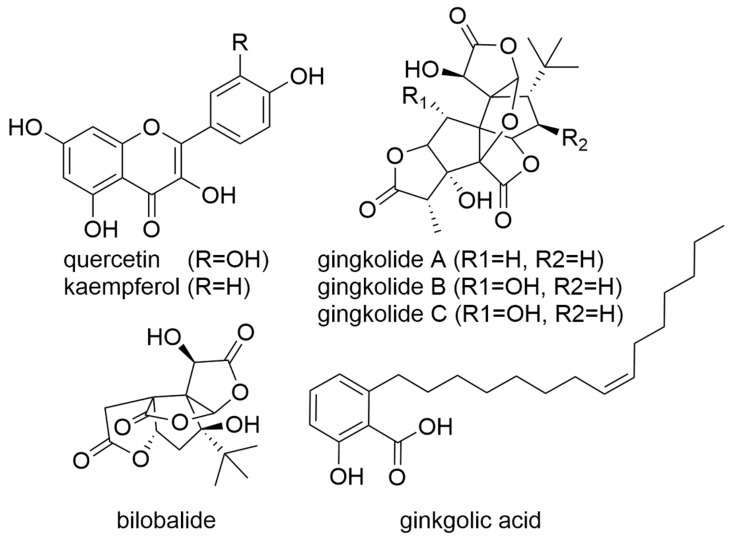
Representative compounds found in *Ginkgo biloba*.

**Figure 4 molecules-27-03209-f004:**
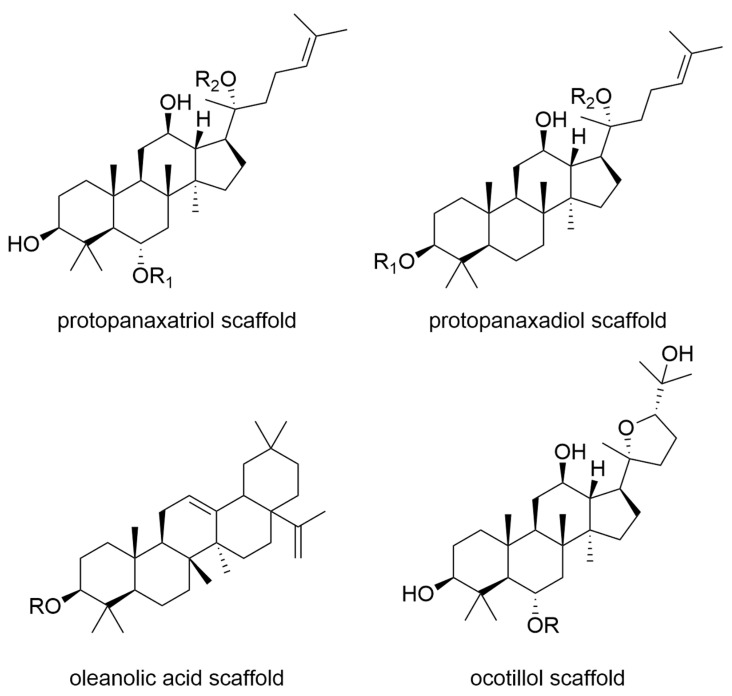
The main aglycone scaffolds from ginsenosides found in *Panax quinquefolius* and *Panax ginseng*.

**Figure 5 molecules-27-03209-f005:**
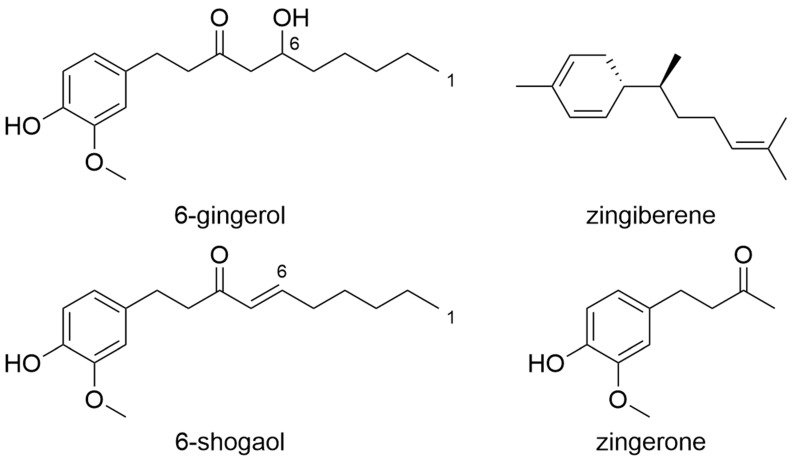
Gingerols and Shogaols from Zingiber officinale (Semwal et al., 2015).

**Figure 6 molecules-27-03209-f006:**
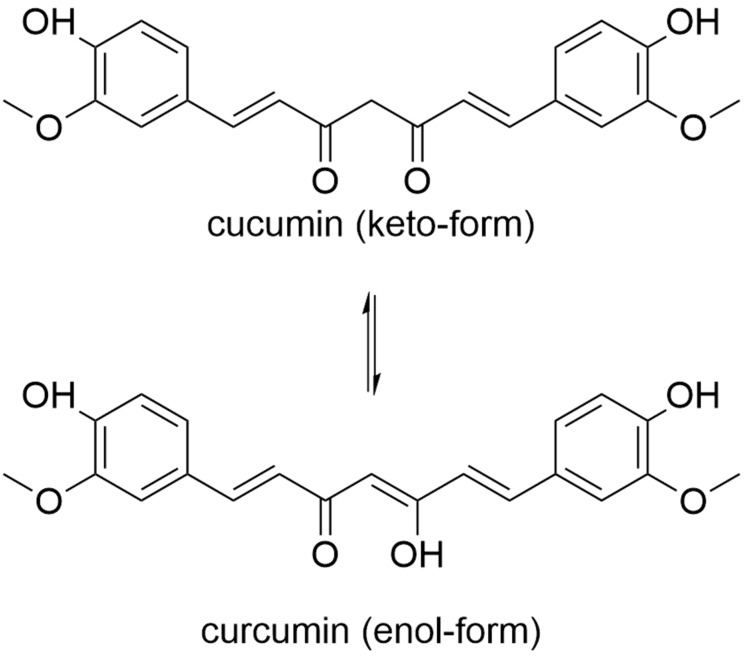
Structure of curcumin in two tautomeric forms.

**Figure 7 molecules-27-03209-f007:**
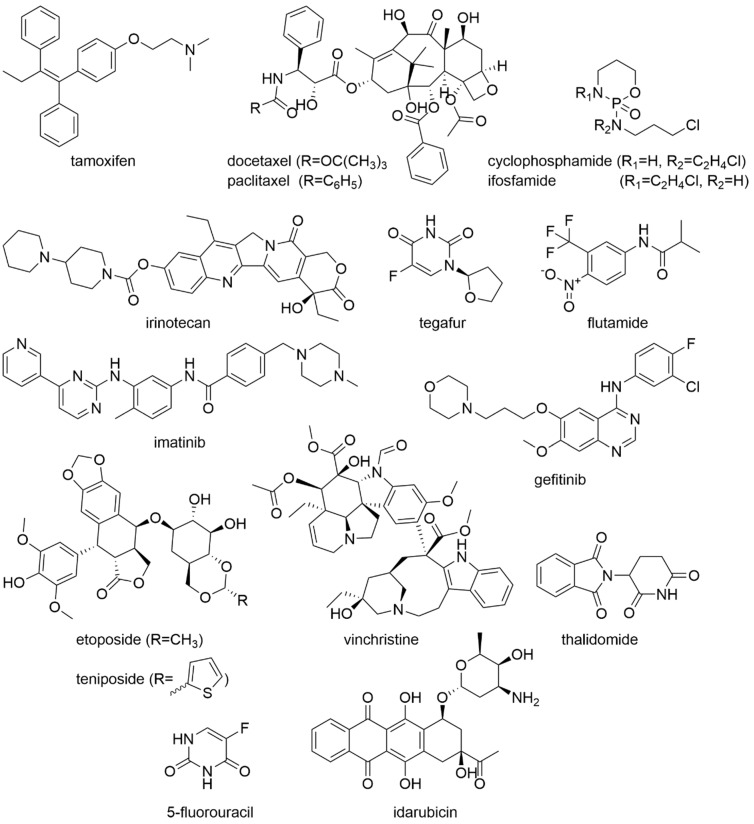
Chemical structures of various drugs potentially affected by the intake of phytopharmaceuticals.

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
