# Peer review of "Phytotherapy in Integrative Oncology—An Update of Promising Treatment Options"

_molecules, 2022, doi:10.3390/molecules27103209_

Round 1

Reviewer 1 Report

This review provides an overview of some important plants and their major components used in integrative oncology and illustrates their pharmacological potential in preclinical and clinical setting. This is interesting and important for phytopharmaceutical studies. However, this manuscript did not organize very well and give as much information as it mentioned in the title/abstract. I suggest that it is back for reconsideration after the following revisions:

- Introduction: Herbal medicines has been used to support human health for thousands of years, but it’s development and application is still limited. The authors should highlight the current research situations of herbal medicines treatment in oncology. For example, the usage and limitations of herbal medicines in different cancer type, population, age range and so on. What is the advantage of phytotherapy compared with other therapy? What’s the limitation? Authors need to leave this clear information throughout the text and discussed in this manuscript.

- Section 2, Basic concepts of phytotherapy: It’s better to integrate this part into the Introduction part. This is also the introduction of phytopharmaceiticals.

- Section 3, Plants and their ingredients in oncology: Why do the authors choose these examples? These plants are important, but not the most important or common ones used for cancer treatment. In addition, not only the ingredients are interested for cancer therapy, but also the molecular mechanism is important. There could be some tables or figures summary the mechanism studies for related ingredients.  

-Figures: Please use the same chemical format for all the figures. Figure 3 and 5 have larger structure and texts than other figures.

- Section 4, Risks, Side Effects and Interactions: Actually, the risk and side effects discussed here are the common problems for phytopharmaceuticals, and already known for many years. I didn’t find any relationship between this part and section 3. Please discuss the limitations of the nature-based ingredients using the ones you discussed in section 3 as examples. Are there some other typical side effects for natural-based compounds studied? The author also can include some clinical data for the risk and side effects. This would be much helpful.

Author Response

Please find the comments attached.

Reviewer 2 Report

Dear Professor,

The manuscript is an important contribution to the development of phytotherapy.  It is a deep, comprehensive study, very well written, focused on an actual, interesting topic. I particularly appreciate the subchapter of “Basic concept of phytotherapy” which differentiates the phytotherapy from other “naturopathic” approaches, and also the subchapter of “Phytopharmaceuticals” which details the regulatory authorities and monographs.   I have no further suggestions, therefore I appreciate the review can be publish in the present form.  

Author Response

Please find the comments attached.

Reviewer 3 Report

Excellent paper, the delight to read! 

The manuscript is well-written and comprehensive. It is based on the analysis of the large amount of literature and is organized in a very logical manner. It can be useful both for the general audience and those researchers who are specialized in the field. 

The only small comment, please consider to replace cancer as one of the leading causes of death rather than the leading cause of death. 

Author Response

Please find the comments attached.

Round 2

Reviewer 1 Report

This review provides an overview of the most important plants and nature-based compounds used in integrative oncology and illustrates their pharmacological potential in preclinical and clinical settings. The revised manuscript clarified the advantage and limitation of phytotherapy, and more details for each compounds. This is very meaningful for researches on nature products. In my opinion, this manuscript now is good for publication on Molecules.